# Genetic Profiling of Malignant Melanoma Arising from an Ovarian Mature Cystic Teratoma: A Case Report

**DOI:** 10.3390/ijms22052436

**Published:** 2021-02-28

**Authors:** Kohei Nakamura, Eriko Aimono, Reika Takamatsu, Shigeki Tanishima, Tomonari Tohyama, Katsutoshi Sasano, Hiroshi Sakuma, Hiroshi Nishihara

**Affiliations:** 1Genomics Unit, Keio Cancer Center, Keio University School of Medicine, 35 Shinanomachi, Shinjukuku, Tokyo 160-8582, Japan; eriko0123@keio.jp (E.A.); reika-t@keio.jp (R.T.); hnishihara1971@keio.jp (H.N.); 2Department of Obstetrics and Gynecology, Kumagaya General Hospital, Saitama 360-8657, Japan; h-sakuma@kumasou.or.jp; 3Department of Biomedical Informatics, Kansai Division, Mitsubishi Space Software Co., Ltd., Tokyo 105-6132, Japan; Tanishima.Shigeki@mss.co.jp; 4Department of Laboratory, Kumagaya General Hospital, Saitama 360-8657, Japan; k-byouri@kumasou.or.jp (T.T.); k-sasano@kumasou.or.jp (K.S.)

**Keywords:** malignant melanoma, mature cystic teratoma, genome sequencing, precision medicine

## Abstract

Ovarian mature cystic teratomas comprise tissues derived from all three germ layers. In rare cases, malignant tumors arise from ovarian mature cystic teratoma. A variety of tumors can arise from mature cystic teratoma, among which primary malignant melanoma (MM), for which no molecular analyses such as genomic sequencing have been reported to date, is exceedingly rare, thereby limiting possible therapeutic options using precision medicine. We used targeted gene sequencing to analyze the status of 160 cancer-related genes in a patient with MM arising from an ovarian mature cystic teratoma (MM-MCT). *KRAS* amplification and homozygous deletion in *PTEN* and *RB1* were detected in tumor samples collected from the patient. No *KRAS* amplification has been previously reported in cutaneous MM, indicating that the carcinogenesis of MM-MCT differs from that of primary cutaneous melanomas. A better understanding of the underlying genetic mechanisms will help clarify the carcinogenesis of MM-MCT. In turn, this will enable treatment with novel targeting agents as well as the initial exploration of gene-based precision oncological therapies, which aim to improve treatment outcomes for patients with this disease.

## 1. Introduction

Malignant melanoma (MM) is the most common primary cutaneous melanoma (PCM), and its molecular and pathological characteristics have been well investigated [1,2]. The key cellular pathways, such as CDKN2A/CDK4/CCND1/RB1, MAPK, and PI3K/AKT, and significant genes, such as *BRAF*, *NRAS*, *TERT*, *TP53*, *NF1*, and *RB1*, are considered to play critical roles in PCM tumorigenesis [3,4,5,6].

MM arising from an ovarian mature cystic teratoma (MCT) (MM-MCT) is extremely rare. To our knowledge, fewer than 40 cases of MM-MCT have been reported to date since the first case was reported in 1901 by Andrews [7,8,9,10], and there have been no reports on its genetic profile. The diagnosis of MM-MCT prior to surgery is impossible. Furthermore, its etiopathogenesis and predictive factors are not understood, and effective treatment methods for such tumors remain elusive due to their paucity.

Precision medicine, using approaches such as comprehensive genome sequencing, is a possible treatment option for several cancers. In particular, recently developed molecular and whole-exome sequencing is being used to reveal the genetic basis of MM-MCT [11].

In the present study, we analyzed 160 cancer-related genes from a patient with MM-MCT. This information will help clarify the carcinogenesis of MM-MCT and will be useful in identifying potential therapeutic targets for future precision medicine approaches.

## 2. Case Presentation

The patient was a 62-year-old woman who underwent right salpingo-oophorectomy based on the preoperative diagnosis of an ovarian MCT. A preoperative laboratory examination showed elevated serum levels of cancer antigen (CA) 19–9 and CA125 (10,159 and 62 U/mL, respectively). There was no family history.

Macroscopically, the ovarian tumor had formed a small sac in a huge cyst that consisted of fibrotic walls and dark brown contents. Small, pigmented nodules were scattered in the stroma of the walls (Figure 1). A microscopic examination revealed that the huge cyst was lined with hemosiderin-laden or foamy macrophages, and there was no lining of epithelial cells (Figure 2). A small sac was lined with squamous-type epithelium and respiratory epithelium (Figure 3). Scattered pigmented nodules (maximum of 14 mm) were composed of the medullary proliferation of anaplastic cells with enlarged nuclei (Figure 4a,b). The immunohistochemical staining of atypical cells revealed a positive result for MelanA (Figure 4c) and HMB45. In addition, high power field observation confirmed the presence of melanin granules that were positive for Fontana-Masson stain (Figure 4d) and negative for colloidal iron stain (Figure 4e). Based on these observations, we made the primary diagnosis of MM accompanied with an MCT and an endometriotic cyst. The gross or microscopic findings alone could not indicate the origin of the melanoma derived from the MCT or endometriotic cyst.

After the surgery, we performed a PET-CT scan and confirmed the diagnosis as MM-MCT. The patient underwent pembrolizumab treatment, which is a standard treatment for PCM. However, the liver metastasis progressed, and the treatment was not effective. Subsequent ipilimumab and radiation therapy to the liver metastasis (20 Gy/5 fr) also failed to control the disease. Two years after the surgery, the patient died of progressive disease.

Genomic DNA was obtained from a sample classified as MM-MCT, but not from mature cystic teratoma and endometriotic cyst, which was thought to be a separate precursor lesion because of the shortage of the sample volume. We performed a cancer gene profiling test by PleSSision-160, as previously described [12,13,14]. The average sequencing depth was 706.5× for MM-MCT. The average tumor cellularity was 70%, as determined histologically.

Several actionable gene alterations were detected in the MM-MCT sample as follows. Homozygous deletion (HD) was detected in both *PTEN* and *RB1* (Figure 5). Oncogene amplification was detected in *KRAS* (estimated copy number: 4.4) (Figure 5). *TSC1* (p.P366Q) and *EPCAM* (p.L286Afs*13) variants were considered as variants of unknown significance. The tumor cells of MM-MCT showed a lack of immunoreactivity for PTEN and RB1, which is consistent with *PTEN*/*RB1* HD status. After analyzing DNA, we examined the results of immunohistochemical staining. The tumor cells of MM were negative for PTEN and RB1 (Figure 6a,b), which is consistent with *PTEN*/*RB1* HD status. Conversely, the epithelium of the MCT showed a mosaic pattern for PTEN and RB1 staining (Figure 7a,b). Regarding the endometriosis lesions, we could not evaluate the expression of PTEN and RB1 as there were no epithelial cells. However, we concluded that the MM had probably arisen from MCT because *PTEN* and *RB1* HD status was observed in both lesions.

## 3. Discussion and Conclusions

MCTs constitute 10–20% of all ovarian neoplasms. They tend to be present in young women around the age of 30 years. MCTs are composed of well-differentiated derivations from at least two of the three germ cell layers. They contain developmentally mature skin, complete with hair follicles and sweat glands, and they occasionally include luxuriant clumps of long hair and pockets of sebum, blood, fat, bone, nails, teeth, eyes, cartilage, and thyroid tissue. MCTs are usually benign but undergo malignant transformation in less than 0.2% of cases, with an incidence of 1–3% [15]. Several malignancies may develop from any of the three germ cell layers, such as adenocarcinoma, malignant thyroid struma, carcinoid tumors, melanomas, and a variety of soft tissue sarcomas [15]. The most common malignant evolution is squamous cell carcinoma from the ectoderm [15]. In a previous study, gene alterations of *TP53*, *PIK3CA,* and *CDKN2A* were frequently observed in squamous cell carcinoma derived from teratoma [16]. Meanwhile, MM-MCTs are extremely rare, with an estimated incidence of <1%; thus, the genomic characters are to be clarified.

We sequenced 160 cancer-related genes (Appendix A) in the tumor sample obtained from the patient diagnosed with MM-MCT and accordingly detected *KRAS* amplification, as well as PTEN and RB1 HD, in the sample. Furthermore, an immunohistochemical analysis revealed that the tumor cells in the MCT area lost the expression of the PTEN and RB1 proteins. Based these observations, we hypothesized that the MCT harbored the same gene alteration in KRAS, PTEN, and RB1, and we speculated that the MM had arisen from the MCT. Additionally, we identified that the endometriosis area did not lose the expression of the PTEN and RB1 proteins, suggesting that endometriosis accidentally coexists with an MM-MCT.

In this case, we focused on the association between origin sites and genetic events. The origin site was an ovarian MCT, an uncommon origin site despite being observed most often in the cutaneous sites.

PTEN is a tumor suppressor gene that is mutated at a high frequency in a wide variety of human cancers (such as glioblastoma, prostate, breast, and osteosarcoma) and was recently identified as a key driver of osteosarcoma in a murine forward genetic screen [17,18]. PTEN encodes a lipid phosphatase that dephosphorylates phosphatidylinositol (3,4,5)-triphosphate (PIP3) to oppose the activity of PI3K, which functions as a PIP3 kinase and is constitutively activated and functions as an oncogene in many cancers [19]. The loss of PTEN causes the persistent activation of the Akt serine–threonine kinase, which promotes cell growth and proliferation, inhibits apoptosis, and controls metabolism by directly phosphorylating numerous downstream targets such as BAD and FOXO and transcription factors such as CASP9, MTOR, and MDM2 [20].

*RB1* is a tumor suppressor gene and is a key regulator of E2F transcription factors and cell cycle progression; thus, its deletion is often accompanied by defects in cell cycle exit and can result in an undifferentiated cellular phenotype [21,22,23]. In addition, pRB controls mesenchymal cell differentiation through interactions with RUNX2 to promote osteogenic differentiation [22] or by blocking PPAR-γ expression to suppress adipogenesis [24]. Though less commonly associated with familial melanoma than with hereditary retinoblastoma, patients with the germline inactivation of RB1 are predisposed to melanoma [25]. However, RB1 HD has not been reported in PCM cases.

Regarding KRAS alterations, KRAS mutations or amplifications have not been described in human melanocytic lesions, whereas NRAS mutations have been correlated with the carcinogenesis of PCM.

In a previous study on patients with metastatic colorectal cancer, PTEN loss and KRAS mutations were shown to have beneficial effects on cetuximab and irinotecan treatment [26]; however, no treatment that was beneficial for all variants (PTEN/RB1/KRAS) was reported.

The recent and growing interest in the unique genetic pathways involved in neoplasia is partially a consequence of the promising developments in targeted therapies, which contribute to the increasing potential of personalized medicine strategies. Previous studies have identified a number of genetic abnormalities associated with oncogenes and tumor suppressor genes in the tumorigenesis of multiple cancer types. Based on the findings of this report, it is possible that PCM carcinogenesis differs from that of MM-MCT. Due to the lack of standard treatment for MM-MCT, our patient often underwent treatments based on immune checkpoint inhibitors, such as pembrolizumab. However, these treatments were not effective. In such cases of rare diseases, we have to consider targeted therapies based on the genetic profile in each case.

This is the first study to report the genomic profiling of MM-MCT. Further genetic analyses of MM-MCTs will undoubtedly help elucidate the mechanisms of the carcinogenesis of PCM and MM-MCTs, as well as provide a basis for novel gene-targeted therapy as a step toward the development of precision medicine strategies.

## Figures and Tables

**Figure 1 ijms-22-02436-f001:**
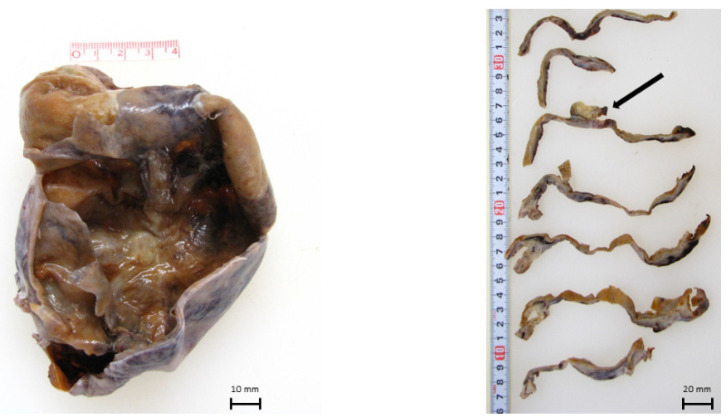
Macroscopic findings: The tumor comprised a small sac (arrow) in a huge cystic legion.

**Figure 2 ijms-22-02436-f002:**
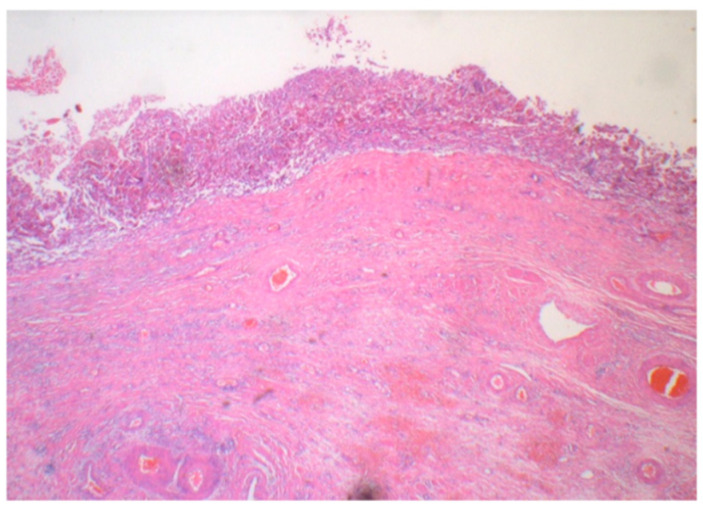
Histologically, the huge cyst was lined with hemosiderin-laden macrophages, and the epithelium was exfoliated (H&E staining) (5×).

**Figure 3 ijms-22-02436-f003:**
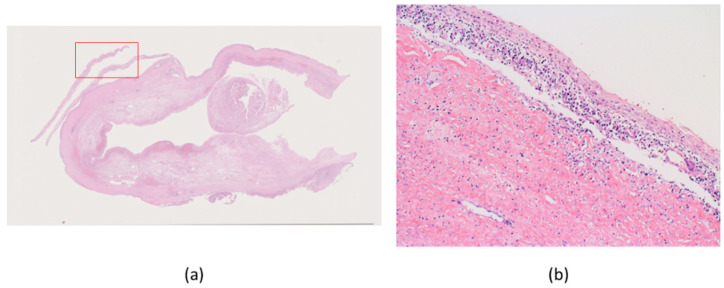
A mature cystic teratoma was observed in a small sac, containing squamous epithelium (low-power: (**a**: 0.4×), high-power: (**b**: 5×)) (H&E staining).

**Figure 4 ijms-22-02436-f004:**
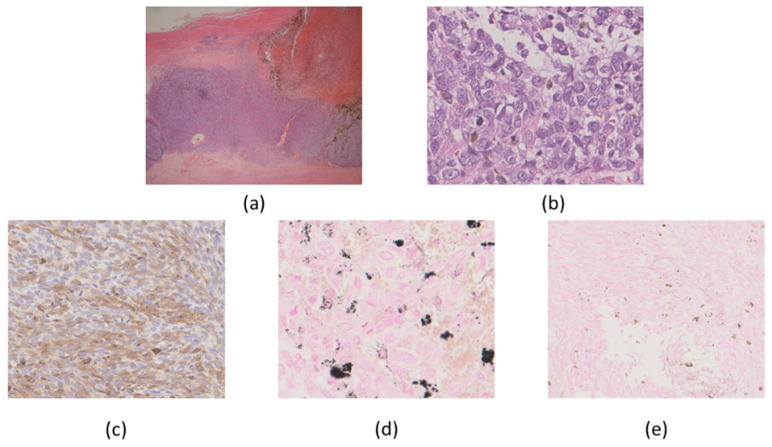
Fibrotic wall showing scattered pigmented areas (**a**: 2×). Tumor cells consisted of enlarged nuclei and melanin granules (**b**: 20×). Immunohistochemically, tumor cells were positive for MelanA (**c**: 10×). The granules were positive for Fontana-Masson (**d**: 20×), and negative for collioid iron staining (**e**: 10×).

**Figure 5 ijms-22-02436-f005:**
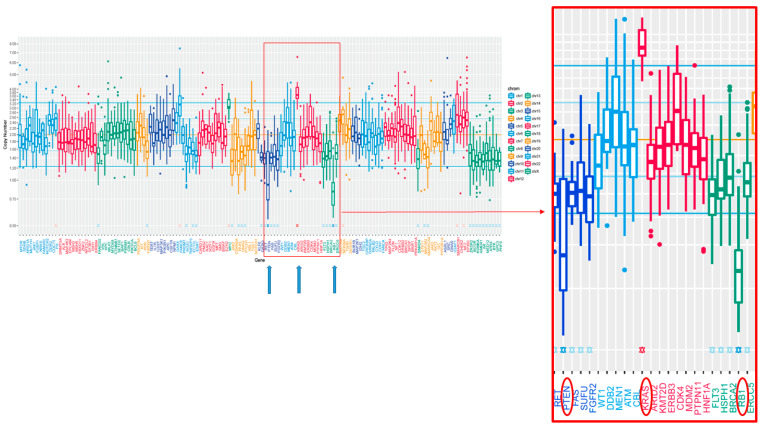
Copy number plot of cancer-related 160 genes. The horizontal axis indicates chromosomal location of the examined genes, and the vertical axis indicates the calculated copy number of each gene. The blue arrow indicated the amplification of *KRAS* and homodeletion of *PTEN* and *RB1*.

**Figure 6 ijms-22-02436-f006:**
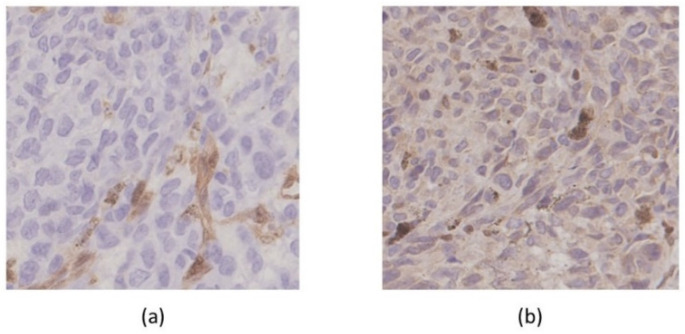
Immunohistochemical findings. Malignant melanoma was negative for PTEN (**a**: 20×) and RB1(**b**: 20×).

**Figure 7 ijms-22-02436-f007:**
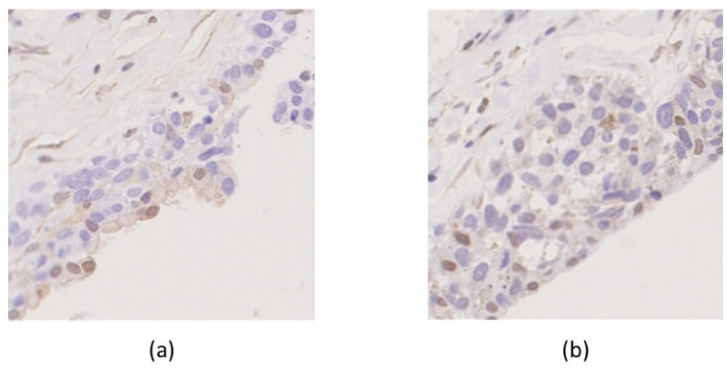
Immunohistochemical findings. The epithelium from mature cystic teratoma showed a mosaic pattern for PTEN (**a**: 10×) and RB1 (**b**: 10×).

## Data Availability

The data presented in this study are available on request from the corresponding author.

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
