# Peer review of "Genetic Profiling of Malignant Melanoma Arising from an Ovarian Mature Cystic Teratoma: A Case Report"

_ijms, 2021, doi:10.3390/ijms22052436_

Round 1
Reviewer 1 Report
In the present manuscript, authors performed a (panel-based sequencing ?) of a rare case of a melanoma that has developed from an ovarian cystic teratoma. Although this case report is interesting the manuscript shows a lack in data supporting the finding.
-absolutely no histological analyses have been performed to validate the melanoma origin of the tumor (MITF, MART1, TYR)
-no Fontana stain has been performed. As there is no pigment/melanon visible, this is a amelanotic melanoma?
-no histological analyses have been performed that the primary "tumor" is indeed a teratoma
-why do authors not present sequensing data in a figure as usual
-have authors validated the KRAS amplification or any of the other mutations?
-The manuscript lacks important references (find attached)
-Did authors check the relevance of their identified mutations in databases like COSMIC

Author Response
We appreciate your excellent comments. Our responses are listed below.
Q1) absolutely no histological analyses have been performed to validate the melanoma origin of the tumor (MITF, MART1, TYR)
A1) We had checked for HMB45 expression, which is a melanoma marker. Based on your suggestion, we have added MelanA staining, which is the most common melanoma marker, and confirmed that the tumor is a melanoma. We have added these findings in the Case presentation section (Please see page 2 line 62-65, Figure 4).
Q2) no Fontana stain has been performed. As there is no pigment/melanon visible, this is a amelanotic melanoma?
A2) We replaced Figure 4b to clarify that melanin pigment is visible. The melanin granules were positive for Masson-Fontana and negative for colloidal iron staining. We added these findings in the Case presentation section (Please see page 2, line 62-65, Figure 4).
Q3) no histological analyses have been performed that the primary "tumor" is indeed a teratoma
A3) The gross or microscopic findings alone could not indicate that the origin of the melanoma was from MCT or endometriotic cyst. We concluded that the primary tumor would be a teratoma by genomic sequencing results. This is stated in the Case presentation, Discussion, and Conclusion sections (Please see page 2, line 66-68).
Q4) why do authors not present sequensing data in a figure as usual
A4) According to your suggestion, we added the copy number plot of 160 cancer-related 160 genes in Figure 5.
Q5) have authors validated the KRAS amplification or any of the other mutations?
A5) According to recent studies, a single round of Sanger sequencing is more likely to incorrectly refute a true positive variant from NGS than to correctly identify a false positive variant from NGS. Validation of NGS-derived variants using Sanger sequencing has limited utility; therefore, we have not validated these gene variants by Sanger sequencing.
Q6) The manuscript lacks important references (find attached)
A6) We have added the recommended citations (Please see page 7, reference no.8-10).
Q7) Did authors check the relevance of their identified mutations in databases like COSMIC
A7) We have checked the pathogenicity of our identified mutations in COSMIC and ClinVar databases.
Reviewer 2 Report
Thank you for the great opportunity to review the manuscript.
This is the case report regarding a malignant melanoma arising from an ovarian mature cystic teratoma. The authors performed an additional analysis of 160 cancer-related genes.
Interestingly, the authors found KRAS amplification and homozygous deletion in PTEN22 and RB1 in analyzed tumor samples collected whereas KRAS amplification was not present.
The paper adds value to the current knowledge about this rare entity. However, I recommend introducing some minor changes in the manuscript.
1. Could you provide more details on the diagnostic process and treatment? Have you excluded the presence of other hamartomas and related genetic syndromes (for example, PTEN hamartoma tumor syndromes)? Was the liver metastases confirmed by biopsy? What does it mean that "the treatment was not effective?" (new distant metastases?). You have also mentioned the use of radiotherapy - what kind of? Palliative? Definitive? What was the prescribed dose and aim of the irradiation?
2. Pembrolizumab is not a standard treatment in cutaneous melanoma, it is one of the possible options in BRAF-negative melanomas (pembrolizumab, nivolumab, nivolumab+ipilimumab).
3. Could you discuss:
- possible therapeutic implications of your findings (in simple words - do we have any drugs/molecules targeted to PTEN/KRAS/RB1),
- genetic profiling and treatment results of other rare subtypes of melanomas (for example mucosal, acral, subungual melanomas),
- genetic profiling of other cancers that arose from MCT (see examples of such case reports in proposed references).
4. "However, these treatments may not be effective because carcinogenesis is different." - I don't think that different carcinogenesis is a reason for the ineffectiveness of immunotherapy in this particular case; it's just the authors' hypothesis. I recommend avoiding such statements.
Useful references (examples):
Dillon LM, Miller TW. Therapeutic targeting of cancers with loss of PTEN function. Curr Drug Targets. 2014 Jan;15(1):65-79. doi: 10.2174/1389450114666140106100909.
McLoughlin NM, Mueller C, Grossmann TN. The Therapeutic Potential of PTEN Modulation: Targeting Strategies from Gene to Protein. Cell Chem Biol. 2018 Jan 18;25(1):19-29. doi: 10.1016/j.chembiol.2017.10.009. Epub 2017 Nov 16.
Zhang X, Sjöblom T. Targeting Loss of Heterozygosity: A Novel Paradigm for Cancer Therapy. Pharmaceuticals (Basel). 2021;14(1):57. Published 2021 Jan 13. doi:10.3390/ph14010057
Uras IZ, Moll HP, Casanova E. Targeting KRAS Mutant Non-Small-Cell Lung Cancer: Past, Present and Future. Int J Mol Sci. 2020;21(12):4325. Published 2020 Jun 17. doi:10.3390/ijms21124325
Gillson J, Ramaswamy Y, Singh G, et al. Small Molecule KRAS Inhibitors: The Future for Targeted Pancreatic Cancer Therapy?. Cancers (Basel). 2020;12(5):1341. Published 2020 May 24. doi:10.3390/cancers12051341
Teterycz P, Czarnecka AM, Indini A, Spałek MJ, Labianca A, Rogala P, Cybulska-Stopa B, Quaglino P, Ricardi U, Badellino S, Szumera-Ciećkiewicz A, Falkowski S, Mandala M, Rutkowski P. Multimodal Treatment of Advanced Mucosal Melanoma in the Era of Modern Immunotherapy. Cancers (Basel). 2020 Oct 26;12(11):3131. doi: 10.3390/cancers12113131. PMID: 33114734; PMCID: PMC7692305.
Newell F, Kong Y, Wilmott JS, et al. Whole-genome landscape of mucosal melanoma reveals diverse drivers and therapeutic targets. Nat Commun. 2019;10(1):3163. Published 2019 Jul 18. doi:10.1038/s41467-019-11107-x
Borkowska A, Szumera-Ciećkiewicz A, Spałek M, et al. Mutation profile of primary subungual melanomas in Caucasians. Oncotarget. 2020;11(25):2404-2413. Published 2020 Jun 23. doi:10.18632/oncotarget.27642
Wield A, Hodeib M, Khan M, Gubernick L, Li AJ, Kandukuri S. Sebaceous carcinoma arising within an ovarian mature cystic teratoma: A case report with discussion of clinical management and genetic evaluation. Gynecol Oncol Rep. 2018 Aug 29;26:37-40. doi: 10.1016/j.gore.2018.08.011. PMID: 30225334; PMCID: PMC6138851.
Cooke SL, Ennis D, Evers L, Dowson S, Chan MY, Paul J, Hirschowitz L, Glasspool RM, Singh N, Bell S, Day E, Kochman A, Wilkinson N, Beer P, Martin S, Millan D, Biankin AV, McNeish IA; Scottish Genomes Partnership. The Driver Mutational Landscape of Ovarian Squamous Cell Carcinomas Arising in Mature Cystic Teratoma. Clin Cancer Res. 2017 Dec 15;23(24):7633-7640. doi: 10.1158/1078-0432.CCR-17-1789. Epub 2017 Sep 27. PMID: 28954785.
Paliogiannis P, Cossu A, Capobianco G, Sini MC, Palomba G, Virdis G, Dessole M, Palmieri G. Squamous cell carcinoma arising in mature cystic teratoma of the ovary: report of two cases with molecular analysis. Eur J Gynaecol Oncol. 2014;35(1):72-6. PMID: 24654467.
Fujii K, Yamashita Y, Yamamoto T, Takahashi K, Hashimoto K, Miyata T, Kawai K, Kikkawa F, Toyokuni S, Nagasaka T. Ovarian mucinous tumors arising from mature cystic teratomas--a molecular genetic approach for understanding the cellular origin. Hum Pathol. 2014 Apr;45(4):717-24. doi: 10.1016/j.humpath.2013.10.031. Epub 2013 Nov 13. PMID: 24485845.
Author Response
We appreciate your critical reading and comments. Our responses are listed below.
Q1) Could you provide more details on the diagnostic process and treatment? Have you excluded the presence of other hamartomas and related genetic syndromes (for example, PTEN hamartoma tumor syndromes)? Was the liver metastases confirmed by biopsy? What does it mean that "the treatment was not effective?" (new distant metastases?). You have also mentioned the use of radiotherapy - what kind of? Palliative? Definitive? What was the prescribed dose and aim of the irradiation?
A1) We performed PET-CT to evaluate whether the melanoma originated from other organs. However, we did not detect tumors in other sites except for metastasis to the liver. Further, there was no family history, including PTEN hamartoma tumor syndromes. Liver metastasis was diagnosed by the PET-CT. "the treatment was not effective" means that the liver metastasis progressed. Radiotherapy for the liver metastasis was definitive, and the dose was 20 Gy/5 fr. We added this information in the Case presentation section (Please see page 2, line 54, and page 4, line 81-85).
Q2) Pembrolizumab is not a standard treatment in cutaneous melanoma, it is one of the possible options in BRAF-negative melanomas (pembrolizumab, nivolumab, nivolumab+ipilimumab).
A2) In Japan, there is no standard treatment for ovarian melanoma. Because this tumor is negative for BRAF, we selected the treatment based on BRAF-negative melanomas (pembrolizumab etc.).
Q3) Could you discuss:
- possible therapeutic implications of your findings (in simple words - do we have any drugs/molecules targeted to PTEN/KRAS/RB1),
- genetic profiling and treatment results of other rare subtypes of melanomas (for example mucosal, acral, subungual melanomas),
- genetic profiling of other cancers that arose from MCT (see examples of such case reports in proposed references).
A3) According to your suggestion, we have included discussion on the suggested topics under the Discussion and Conclusion sections (Please see page 5, line 128-130, and page 6, line 165-168).
Q4) "However, these treatments may not be effective because carcinogenesis is different." - I don't think that different carcinogenesis is a reason for the ineffectiveness of immunotherapy in this particular case; it's just the authors' hypothesis. I recommend avoiding such statements.
A4) Your comment is valid. Based on your suggestion, we have deleted this sentence.
Round 2
Reviewer 1 Report
In the revised version of the manuscript, authors provide more information about the characteristics of the cellular subset found in the teratoma and improved the manuscript. However, I strongly suggest that authors choose another way to present stainings in figures 2-4 and either show a higher magnification of figure 5 and or show a magnification of important genes shonwn...At least neither the laneling nor the legends are readable. This must be improved prior to publication.
Author Response
We appreciate your excellent comments.
We replaced Figure 5 with a magnification of important genes to make it to be readable. Additionally, we added Figure 3 with image of low-power field.